# Lyophilization of Nanoparticles, Does It Really Work? Overview of the Current Status and Challenges

**DOI:** 10.3390/ijms241814041

**Published:** 2023-09-13

**Authors:** Matthew S. Gatto, Wided Najahi-Missaoui

**Affiliations:** Department of Pharmaceutical and Biomedical Sciences, College of Pharmacy, University of Georgia, Athens, GA 30602, USA; matthew.gatto@uga.edu

**Keywords:** nanoparticles, stability, lyophilization, freeze-drying, lyoprotectants

## Abstract

Nanoparticles are being increasingly used as drug delivery systems to enhance the delivery to and uptake by target cells and to reduce off-target toxicity of free drugs. However, although the advantages of nanoparticles as drug carriers are clear, there are still some limitations, especially in maintaining their long-term stability. Lyophilization, also known as freeze-drying, has been heavily investigated as a solution to this problem. This strategy has been shown to be effective in increasing both the long-term stability of nanoparticles and the shelf life of the drug product. However, the process is still in need of improvement in several aspects, such as the process parameters, formulation factors, and characterization techniques. This review summarizes the advantages and limitations of nanoparticles for the treatment of disease, advantages and limitations, and the status of the lyophilization of nanoparticles for therapeutic use and provides insight into both the advantages and the limitations.

## 1. Introduction

Safe and effective drug development is at the forefront of providing effective healthcare to patients in need of therapeutic treatment. The drug development process can take as long as 10–15 years and cost well over USD 1 billion, yet despite this massive investment by pharmaceutical companies, about 90% of drugs in the development stage never make it to the market [1]. In an effort to avoid lost investment and bring effective treatments to the public, novel drug delivery methods have been a primary research focus within pharmaceutical and biomedical sciences. This research includes nanotechnology, which employs nanoparticles on a 1–100 nanometer scale. These particles are intended to perform a variety of functions, such as acting as biosensors, assisting in imaging, and delivering drugs [2,3].

Nanoparticles as a drug delivery system are still very much in their infancy. However, nanotechnology offers a promising solution to many of the clinical shortcomings most pipeline drugs experience during development. These nanoparticles possess remarkable physiochemical and biological properties that are highly advantageous to drug delivery. They have been shown to increase the solubility of a drug, decrease the toxicity of the treatment, enhance the specific delivery of the drug, and control the rate of drug release [4]. However, although the advantages of nanotechnology as drug carriers are clear, there are still some uncertainties about maintaining the stability of nanoparticles for long enough to consider them therapeutically relevant [5]. Therefore, there exists a need for a strategy to preserve nanoparticle efficacy over a long period of time.

The lyophilization of nanoparticles has been heavily investigated as a solution to this problem. Lyophilization, also known as freeze-drying, utilizes the sublimation of ice at low pressure to remove water from a sample [6,7]. This strategy has been shown to be effective in increasing both the long-term stability of nanoparticles and the shelf life of the drug product [7]. However, the process is far from perfect today, as the process parameters, formulation factors, and characterization techniques still need refinement. This review aims to describe the current status of the lyophilization of nanoparticles for therapeutic use and provide insight into both the advantages and the limitations.

## 2. Nanoparticles in Drug Delivery

### 2.1. Current Status of Nanoparticle Therapeutics

The research into nanoparticles for therapeutic use does not have an extended history, as the first drug formulations were not approved by the FDA until the 1990s [8]. However, as their potential has been recognized through clinical studies over the years, an increasing number of pharmaceutical companies have turned their attention towards nanoparticle-based treatments. As of 2022, there have been around 100 additional nanomedicines brought to the market, as well as 563 formulations that are still in clinical trials [9]. Furthermore, the market value of nanomedicine is projected to grow from USD 53 billion in 2009 to USD 334 billion by 2025 [10]. This shows the intense investment that nanomedicine is receiving in today’s healthcare landscape.

The ability to easily manipulate nanoparticles has led to the development of treatments for a wide range of diseases. The majority (53%) of nanomedicines’ indications are for treating cancers [11], as seen in Figure 1. This comes as no surprise, as cancer is, on average, the costliest health condition in the United States [12], and pharmaceutical companies have been looking for delivery methods to increase both the efficacy and the efficiency of cancer treatments. Nanoparticles have been a prime focus in this regard.

In addition to a wide range of indications, nanomedicines have also taken on a wide range of forms. The majority of approved nanoparticle therapeutics include polymer-based formulations (35%), lipid-based formulations (29%), and nanocrystalline formulations (10%) (Figure 2) [13]. Notably, however, the entire list of approved or in-development nanodrugs includes more than 14 different nanoparticle types [12], highlighting the multitude of possibilities nanotechnology can offer in drug delivery.

### 2.2. Advantages of Nanoparticles for Drug Delivery

#### 2.2.1. Solubility

The insolubility of the active pharmaceutical ingredient (API) in water is a major barrier in traditional drug delivery. Because roughly half of human blood is water, water-insolubility will prevent the free movement of a drug through the bloodstream in vivo. Despite the importance of this physiochemical property, it has been estimated that about 40% of currently approved drugs and 90% of in-development drugs are poorly water-soluble [14].

Nanoparticles have the ability to increase the solubility of drug substances. Many nanoparticles, such as micelles, liposomes, and dendrimers, possess internal cores or pockets that can favorably house a compatible drug, as well as a hydrophilic exterior that allows the vehicle as a whole to be solubilized in the blood stream [15]. The encapsulation of the API within the nanoparticle causes the drug to take on the solubility properties of the carrier. As a result, a poorly water-soluble drug can travel through the bloodstream as though it were water-soluble. Additionally, therapeutic nanoparticles can also be engineered with surface modifications that increase their water solubility even further [16].

#### 2.2.2. Specificity

Most traditional chemotherapy drugs are cytotoxic by nature and aim to kill cancer-causing cells. However, these drugs often display cytotoxicity towards healthy cells as well, resulting in the debilitating side effects that are seen in most patients under chemotherapy [17]. Nanoparticles are being increasingly investigated as potential drug carriers for chemotherapeutic agents. This is largely because of the unique ability to manipulate their structures and their surfaces to enhance specificity and drug uptake by cancer cells. Ligands, such as antibodies or nucleic acids, that are specific to receptors on tumor cells can be conjugated to the surface of nanoparticles, resulting in targeted delivery [18]. For example, human epidermal growth factor receptor-2 (HER-2) is a cell-membrane protein responsible for promoting signals related to cellular proliferation and differentiation [19]. HER-2 is overexpressed in breast cancer, leading to uncontrolled tumor growth [19]. Therefore, an anticancer drug, like docetaxel, can be encapsulated in a nanoparticle and conjugated with an antibody for HER-2 to achieve targeted cancer therapy.

It has been shown that drugs encapsulated in nanoparticles with a surface antibody for HER-2 possess an increased therapeutic effect over both the free drug and untargeted nanoparticles [20]. This result shows that nano-encapsulated drug formulations with attached ligands can increase both the specificity and the efficacy of a treatment.

#### 2.2.3. Controlled Release

Nanoparticles are naturally removed from the bloodstream by the immune system, as the body identifies them as foreign elements that need to be neutralized and eliminated. However, it is possible to coat the surface of nanoparticles with inert molecules that reduce interactions with the bloodstream. As a result, it is less likely that an immunogenic response will be triggered. The most commonly used particle of this nature is polyethylene glycol (PEG), and it has been shown to increase the circulation time of a drug, as was shown in many studies that started in the 1970s [21]. Adding PEG to nanoparticles also results in a greater amount of bioavailable drug and a longer duration of action. In a PEGylated liposomal product known as Doxil^®^, studies have shown that the area under the plasma concentration–time curve (AUC) was at least 60 times greater than that of the free drug several hours after injection [22]. Additionally, PEGylated liposomes demonstrated a half-life of up to 50 h, whereas traditional non-PEGylated liposomes only possess a half-life of less than 30 min [22,23].

In addition to PEGylation, nanoparticles can also be prompted to release their drug cargo based on external cues such as temperature or pH. This can be advantageous when the microenvironment of the target tissue has specific characteristics. For example, inflamed tissues as well as tumor tissues are often characterized by a low pH. It has been shown that nanoparticles can be modified to be sensitive to a low pH and, as a result, they release their drug at the desired site only when the pH is below a certain threshold [24]. This ability provides the opportunity for specifically engineered design mechanisms, and it highlights the utility of nanoparticles for drug delivery.

#### 2.2.4. Toxicity

Many effective drugs are limited in use because their toxicity towards healthy cells prohibits the use of high doses and requires that drug doses be minimized during the treatment of disease. One strategy for overcoming this barrier is increasing the specificity of the drug, as described previously. An additional solution is decreasing the off-target toxicity of the drug in the body, and this can be accomplished through the use of nanoparticles. There are numerous studies showing the decreased toxicity of a nano-encapsulated drug as compared to its free-drug form [25,26,27]. For example, He et al. showed that encapsulating teniposide into an albumin nanoparticle resulted in a significantly less toxic and more effective therapy. Incredibly, the observed decrease in white blood cells that resulted from the nano-encapsulated form was only one-third that of the free drug form [28]. This demonstrates the increased safety profile of the nanoparticle treatment.

### 2.3. Different Types of Nanoparticles

It is important to remember that the term “nanoparticle” refers to a broad collection of particles with a size that falls into the 1–100 nm range [2]. As mentioned previously, there are more than a dozen nanoparticles that have been approved by the FDA and more are still in development. They are often organized into groups based on their common functionality or structure. This review focuses on the following three classifications of nanoparticles: lipid-based, polymer-based, and inorganic. These groups were selected based on their similar characteristics as well as their relevance to the treatment of disease. Several other major nanoparticles, like quantum dots, are well researched but contribute most to other fields of medicine, such as imaging and diagnostics. With that said, the major nanoparticles in each group recognized by this review are described briefly below (Figure 3).

#### 2.3.1. Lipid-Based Nanoparticles

##### Liposomes

Liposomes are spherical nanovesicles that are comprised of a hydrophilic center encapsulated by a phospholipid bilayer. The bilayer usually contains cholesterol in levels as high as 66% by mole in order to add structure and stability to the membrane [29]. The phospholipids themselves can be sourced from a wide range of both synthetic or naturally occurring molecules, and each different phospholipid may be better suited for a specific purpose [30]. Additionally, the presence of both an aqueous center and a hydrophobic bilayer means that liposomes are suitable drug carriers for hydrophilic, hydrophobic, and amphiphilic drugs [31,32]. The overall structure of liposomes is quite similar to that of human cell membranes, which makes them biocompatible and biodegradable. In addition, liposomes can be manipulated in many ways that give them an advantage over traditional drug delivery forms [31,33].

##### Exosomes

Exosomes are structured similarly to liposomes; they are also spherical vesicles with a lipid bilayer membrane [34]. However, exosomes differ from liposomes in a few key areas. First, exosome bilayers are composed of non-lamellar-forming lipids, which leads to further stability in the curvature of the membrane [35,36]. Additionally, exosomes are more asymmetrical and contain more membrane proteins than liposomes, both of which are advantageous to the functionality of the vesicle [36]. The human body is no stranger to exosomes; they are secreted naturally by most living cells, and they often carry a biological cargo from the donor cell, such as proteins, RNA, DNA, or lipids, that have a natural therapeutic effect within the body [37]. In other words, nature has been using exosomes as drug carriers for far longer than we have. Nonetheless, exosomes’ low immunogenicity and high stability in human blood has made them an intriguing option for designed drug delivery, and they are receiving increased attention in the scientific community every year despite there being zero FDA-approved exosome therapies currently available [38].

##### Solid Lipid Nanoparticles

Solid lipid nanoparticles (SLNs) take on a different structure from the previous two lipid-based drug carriers. SLNs consist of a solid lipid core containing a dispersed or dissolved active ingredient. The solid core is coated with an emulsifier (stabilizing agent), often in the form of a surfactant [39]. This composition results in a highly biocompatible and biodegradable carrier [40], which are two key criteria in designing an effective drug delivery system. Additionally, SLNs have been shown to be both economical [41] and easily scalable [42], overcoming the major obstacles of liposomes and exosomes, respectively. However, SLNs also come with drawbacks concerning the encapsulated drug substance. Under environmental stresses such as light and temperature, gelation of the particles has been shown to occur [43], which could lead to a loss of drug substance. Furthermore, it has been reported that these nanoparticles already have a low drug-loading capacity as a result of the small structure, leaving little room for error during production [44]. Yet, even with these disadvantages, SLNs remain an intriguing drug delivery system due to their high functionality.

#### 2.3.2. Polymer-Based Nanoparticles

##### Polymeric Nanoparticles

Polymeric nanoparticles (PNPs) utilize natural or synthetic polymers to immobilize a drug within a defined structure for enhanced drug delivery. They can be divided into two groups based on their structure: nanocapsules and nanospheres. Nanocapsules contain a drug dissolved or dispersed in an oil-based solvent encapsulated by a polymeric shell, whereas a nanosphere has an evenly distributed drug–polymer matrix [45,46]. The active component’s release is dictated by a combination of diffusion through the polymer and the degradation of the polymer itself [47]. As a result, a wide variety of polymers is used in order to achieve the desired release conditions. However, the most commonly used polymers for drug delivery are polylactic acid (PLA), polyglycolic acid (PGA), and polylactic-co-glycolic acid copolymer (PLGA) [48]. These three polymers have been approved by the FDA for drug delivery due to their demonstrated safety, biodegradability, low immunogenicity/toxicity, and manipulability. This regulatory track record with the FDA, along with proven increased pharmacokinetic properties [49], have made PNPs gain increased attention from researchers to investigate them as drug delivery systems.

##### Dendrimers

Dendrimers are symmetrical, branched, 3D structures made from repeated structural polymers. There are three main components to dendrimers [50]. The first component is a central core consisting of one atom or molecule that connects all of the polymers [51]. The second component is the collection of branched polymers that repeat concentrically and form a globular structure [51]. The third component is the multitude of surface functional groups that give the dendrimer the flexibility to bind to specific receptors and direct the therapy to its intended target [52]. This unique, branching structure of dendrimers allows for drug substances to be encapsulated in the empty void space in between the polymer branches. Dendrimers share many of the same advantages as other nanoparticles, but their exceedingly amenable design has led to them becoming a major candidate for drug delivery systems, and as a result, there are several dendrimer-based drug products available and many more in clinical trials [53].

##### Micelles

Micelles are nanoparticles with a shell–core structure made of amphiphilic block copolymers. These copolymers align themselves in such a way that the hydrophobic ends of the polymer chains create a lipophilic core, and the hydrophilic ends of the polymer chains create a hydrophilic shell [54]. This self-assembled formation allows micelles to carry hydrophobic drugs as cargo in their core and increase their bioavailability [55,56]. Many characteristics of micelles, such as surface composition and polymer length, can be modified in order to induce a specific change in functionality. For example, similar to other nanoparticles, specific ligands can be added to their surface to increase the selectivity of the micelles’ binding [57]. Additionally, the stability of the micelle can be increased by lengthening the hydrophobic portion of the polymers [58].

Micelles offer a tempting list of advantages for drug delivery; however, the stability of micelles in biological conditions is a major concern. It has been shown that micelles are relatively unaffected in mouse albumin, but they readily destabilize in human albumin, a major component of our blood [58]. This leads to a necessary level of skepticism during preclinical animal studies, as an encouraging result in mouse models still may not translate well to human models. There is much work to be done in this respect to assuage these apprehensions.

#### 2.3.3. Inorganic Nanoparticles

##### Gold Nanoparticles

Gold nanoparticles (Au NPs) are most commonly produced from colloidal synthesis, and they come in several different structures, such as spheres, nanorods, and nanocages [59]. Like other nanoparticles, Au NPs offer many advantages in drug delivery. They can be conjugated with other molecules such as PEG, antibodies, or oligonucleotides to achieve a specific function or delivery mechanism [60]. Furthermore, Au NPs can be activated using near infrared (NIR) light to induce a temperature change that causes the release of a conjugated drug [61]. This unique mechanism of controlled release gives Au NPs a potential advantage over organic nanoparticles for drug delivery. Despite these advantages, there are some potential limitations to Au NPs. There is evidence that these nanoparticles can accumulate in toxic concentrations in the liver or spleen [62], so there are still questions surrounding their biocompatibility.

##### Carbon Nanotubes

Carbon nanotubes are rolled sheets of graphene that form a hollow cylindrical tube [63]. They can come as single-walled carbon nanotubes (SWCNT) or multi-walled carbon nanotubes (MWCNT) [64], both of which are depicted in Figure 3. SWNCTs typically have a diameter of 1–2 nm and a length of up to 100 nm, whereas MWCNTs have a diameter of anywhere from 5 to 100 nm with a length of up to 15,000 nm [65]. The exceedingly long length as compared to the diameter gives carbon nanotubes a high drug-loading capacity, and they are innately capable of passing through biological membranes using several different mechanisms [66]. Additionally, carbon nanotubes can be functionalized using surface modifications that can enhance solubility, improve biocompatibility, and reduce toxicity [67]. However, the high production costs and lack of consistent quality [68] inhibit carbon nanotubes from becoming a dominant carrier in nanotherapeutics for now.

## 3. Lyophilization Process of Nanoparticle Formulations

Lyophilization has emerged as a potential solution to overcome the stability of nanoparticles and has been the focus of many research studies. There are three main steps during lyophilization. These steps are explained below and visualized in Figure 4.

### 3.1. Pre-Freezing

The first stage of lyophilization is the freezing stage. The drug solution is cooled so that ice crystals begin to form. As more water is converted into ice, the drug solute becomes more concentrated in the remaining water, and the viscosity of the solution rapidly increases. This increase in viscosity prevents additional ice crystals from forming, and it is said that the system has entered a glass state. The temperature at which the system reaches this state is known as the glass transition temperature (Tg’) [69].

### 3.2. Primary Drying

The second stage of lyophilization is primary drying. During this stage, the pressure is dramatically reduced to roughly one ten-thousandth of atmospheric pressure [58]. At this low pressure, heat can be added to the system to induce sublimation, the instantaneous conversion of ice to water vapor. The result is a dry, porous system characterized by very high concentrations of the solute in a miniscule amount of remaining water [69].

### 3.3. Secondary Drying

The third stage, known as the secondary drying, involves drying off the remainder of the water in the system that did not convert to ice crystals during the first stage. The system is slowly given additional heat while the pressure remains extremely low. The water is desorbed, and the resulting product is a dried “cake” that exhibits higher stability than the previous liquid form [69].

## 4. The Need for Lyophilization

Nanoparticles offer many advantages in drug delivery, but several obstacles still exist that prevent these carriers from being widely used. This includes challenges such as regulatory guidelines, high cost of production, lack of thorough understanding, and potential toxicity. However, perhaps the biggest obstacle facing the development of nanoparticle therapeutics is the issue surrounding long-term stability. Nanoparticles often experience both physical and chemical instabilities that limit their feasibility and popularity in drug development [70].

Physical instabilities are a common problem for nanoparticles in drug delivery. The high surface area-to-volume ratio creates high free energy on the surface of the particle. As a result, the surface molecules interact with neighboring particles in an attempt to decrease free energy and increase thermodynamic stability [71]. This leads to troublesome phenomena such as aggregation and particle fusion [72]. Aggregation occurs when several particles cluster together and form a group, whereas fusion occurs when multiple nanoparticles interact and combine to form one larger particle. Both instances can lead to drug leakage and loss of function [73].

Chemical instabilities also present a host of problems for nanoparticles. The most commonly observed chemical instability in nanoparticles is the hydrolysis of the polymer membrane [71]. Nanoparticles are often constructed with hydrolytic biodegradable polymers due to their preexisting FDA approval. However, although this minimizes regulatory obstacles, it can lead to the potential hydrolysis and degradation of the nanoparticle membrane [74]. This can obviously initiate drug leakage and render the nanoparticle no longer therapeutically useful. Furthermore, although it was previously mentioned that nanoparticles are often conjugated with ligands on their surface for specific delivery or controlled release, these accessories can also face degradation over time [75]. Thermosensitive ligands added for a specific purpose can face destabilization when faced with external stressors [71]. This loss of function could negatively impact the engineered release mechanism that makes the nanoparticle effective in the first place. In addition to the chemical instabilities of the nanoparticle shell, it is possible for chemical instabilities to exist within the encapsulated drug substance itself [76]. Metabolic instabilities of the encapsulated drug are not completely eliminated by utilizing a protective vehicle. Even if the nanoparticle can successfully deliver the active compound to the desired site, the drug is of no therapeutic benefit if it has degraded over the course of product storage and transportation.

For these reasons, further steps should be taken to ensure the long-term stability of both the nanoparticle and its packaged drug. This is why lyophilization has been commonly utilized. It offers a solution to both the innate and the external instabilities that arise during the use of nanotherapeutics.

## 5. Characterization of Lyophilized Nanoparticles

The production of stable nanoparticles can be a complex protocol in and of itself. When coupling the nanoparticle production process with a potentially destabilizing operation such as lyophilization, it is absolutely vital to ensure that the particles are well characterized at different stages of production. These data are needed to ensure a safe, consistent, and efficacious drug product. Consequently, this review finds it important to describe several different properties that are commonly used to characterize nanoparticles before and after lyophilization.

### 5.1. Size and Size Distribution

When working with something on the nano scale, it should come as no surprise that size is perhaps the most important property to characterize. Particles should be in the 1–100 nm range to be considered nanoparticles [2], yet even at this scale, a small disparity in size can correspond to a massive difference in properties. For example, smaller nanoparticles have a larger surface area-to-volume ratio, meaning that more of the encapsulated drug is in contact with the porous particle surface. As a result, the drug is more likely to diffuse out of the nanoparticle, leading to a faster release and quicker therapeutic effect [77]. However, smaller particles are also associated with higher surface energy levels. Therefore, the particles are more likely to aggregate during storage to reach a more favorable thermodynamic state [77]. On the other hand, particles smaller than 10 nm in diameter are likely to be eliminated by the kidneys over time rather than reach their target tissue [78], yet large nanoparticles (>150 nm) are known to fall victim to filtration in the spleen [79].

The work of Sun et al. shows specifically how a small difference in nanoparticle size can lead to a large difference in pharmacokinetic properties. One hour after administering equivalent doses of 24 and 37 nm nanoparticles in vivo, 50% of the 24 nm nanoparticles were still in circulation compared to just 5% of the 37 nm nanoparticles [80]. These findings suggest that there is little room for error when it comes to producing an optimally effective nanoparticle for drug delivery.

Particle size is most frequently measured using dynamic light scattering (DLS). DLS works by utilizing a photon detector to measure the fluctuations in the intensity of scattered light after beaming a sample with a laser. From the raw data collected, the Brownian diffusion of the particles over time is related to the particle size by the Stokes–Einstein equation. This method is advantageous over traditional electron microscopy because it can be used to measure samples still in solution or suspensions, and it is sensitive to soft particles such as polymers and proteins [81].

Furthermore, DLS can also provide a measure of size distribution in the sample. With respect to nanoparticles, size distribution is often represented with a dimensionless parameter, known as the polydispersity index (PDI), on a scale of 0–1, where a value closer to zero translates to a higher degree of uniformity. In general, a PDI of lower than 0.2 for polymeric nanoparticles or 0.3 for lipid-based nanoparticles is considered acceptable [82]. PDI is paramount to the production of nanoparticles for drug delivery. A marketed drug product not only needs to be safe and efficacious but also needs to be consistent across batches. Characterizing size distribution aids in confirming quality from batch to batch.

With the significant impact that size and PDI can have on the utility of nanoparticles, it is important that these properties stay constant before and after the process of lyophilization. Very little change in these values after lyophilization is a good indicator of a successful drug formulation [7].

### 5.2. Shape

The shape of a nanoparticle can have a great impact on several properties, such as cellular uptake, circulation time, and surface-binding tendency [83]. For example, the cellular uptake of spherical and rod-shaped gold nanoparticles of similar sizes was investigated, and it was found that the spherical nanoparticles experienced a 500% higher rate of uptake than the rod-shaped nanoparticles [84]. In contrast, a 2007 study investigated the circulation time of differently shaped micelles administered in vivo. The results showed that 50% of filamentous micelles remained in circulation for 7 days, whereas spherical micelles of similar volume only remained in circulation for about 48 h [85]. These two examples show conflicting results in terms of the optimal shape for drug delivery. This discrepancy highlights the opportunity for creative problem solving that is evident across all areas of nanoparticles.

Nanoparticle shape is typically observed using high-resolution electron microscopy. Scanning electron microscopy (SEM) is utilized to visualize a 3D representation of the surface topography and morphology of the freeze-dried matrix [86]. This information could be useful in revealing signs of collapse during lyophilization [87]. Another tool used to visualize the shape of nanoparticles is transmission electron microscopy (TEM). TEM can be used to generate a higher-magnification 2D image of the particle shape and features [32,71]. Both forms of microscopy are useful in characterizing the overall structure of the nanoparticle.

### 5.3. Surface Charge

The surface charge of a nanoparticle is represented by a value known as the zeta potential. Zeta potential values can be negative, positive, or zero, but a zeta potential with a greater absolute value indicates a more stable colloidal solution of nanoparticles [88]. This can be explained by the presence of electrostatic repulsions that prevent the particles from interacting with each other and agglomerating [89]. Additionally, zeta potential has been shown to play a major role in determining the tendency of the nanoparticle to attach to a target cell’s membrane. Cell membranes are typically populated by negatively charged molecules [90], and as a result, positively charged nanoparticles are more likely to bind to the cell surface.

Zeta potential is commonly measured by placing the colloidal solution in a chamber with two electrodes. When opposite charges are applied to each electrode, the nanoparticles move toward the electrode with the opposite charge. DLS is then used to relate the velocity of the particles to their charge [91,92]. Monitoring zeta potential before and after lyophilization is crucial in determining the success of the freezing process. Any free drug remaining in the solution from the loading process may decrease the absolute value of the zeta potential during lyophilization [74]. This may lead to the aggregation of nanoparticles during freezing, which would be detrimental to the quality of the final product.

### 5.4. Drug Retention

The retention of the encapsulated drug is obviously important to characterize after lyophilization. Because of all of the physical instabilities that may occur during lyophilization, such as agglomeration or system collapse, there are many opportunities for drug leakage to occur. When developing a drug product, it is crucial both economically and therapeutically that the nanoparticles retain as much of their housed drug as possible [93,94]. A popular method of measuring drug content within nanoparticles is using high-performance liquid chromatography (HPLC). HPLC is capable of both identifying and quantifying the molecules in a sample, making it an ideal choice for measuring the remaining drug in a sample [95,96].

## 6. Factors Affecting the Lyophilization of Nanoparticles

Lyophilization is a complex process with a wide variety of possible parameters and formulation factors. The success of the operation hinges on many different conditions, and the optimization of these conditions is still being investigated. This review will describe four of the major factors that should be considered when utilizing the lyophilization of nanoparticles.

### 6.1. Lyoprotectants

During lyophilization, several stressors threaten the integrity of the nanoparticles in solution. High levels of interaction between the nanoparticles, concentrated pockets of solute and free drug, and mechanical stresses from crystallized ice all present real dangers to the stability of the nanoparticles [97]. Therefore, it is important that the appropriate levels of excipients are added to the sample to protect the nanoparticles from these stresses. Among these excipients are cryoprotectants, which shield the sample from freezing stresses, and lyoprotectants, which shield the sample from drying stresses. Typically, an excipient that falls into one of these categories will also fall into the other [7], so this review will only refer to both of them as lyoprotectants.

The cryoprotective effect is theorized to come from a glassy matrix formed by the excipients at low temperatures that surrounds the nanoparticles and prevents them from interacting with each other [98]. Without interacting, it is impossible for the nanoparticles to aggregate, fuse, or participate in any other destabilizing phenomena with each other. Additionally, this matrix shield provides a buffer from the mechanical stresses of ice crystallization.

The lyoprotective effect, on the other hand, may be explained by the water replacement theory [99]. As water and ice are dried off during lyophilization, the gaps they leave behind within the nanoparticle membrane are structural weak points that could lead to damage. The lyoprotectants replace the water and form hydrogen bonds with the polar end of the membrane, providing rigidity to the nanoparticle once again (Figure 5).

There is a wide selection of lyoprotectants that are commonly used for the lyophilization of nanoparticles. The most popular are typically disaccharides, namely trehalose and sucrose. While both are heavily used, trehalose seems to be accepted as the optimal choice of lyoprotectant. Trehalose is preferred for a couple of reasons. Firstly, it has been reported that a superior protective effect may exist due to its stronger interactions with the membrane of nanoparticles [100]. There is also evidence that suggests that, under conditions of excess water exposure during lyophilization, trehalose may form a dihydrate [101]. This would act as another layer of protection for the nanoparticles from excess water penetration [102]. A recent study highlighted the difference in the protective effect of trehalose versus the protective effect of sucrose. In studies conducted by Luo et al. [103], liposomes were freeze dried using either sucrose or trehalose as a lyoprotectant. Then, the nanoparticles were stored at 40 degrees Celsius for a month, and their resulting particle size was compared to the initial size. When sucrose was used as a lyoprotectant, the particle size increased about 8-fold on average. When trehalose was used, the same measure increased less than 2-fold on average. A similar trend was observed for solid lipid nanoparticles (SLNs) as well [103]. This shows the drastic effect that an individual lyoprotectant can have.

In addition to sucrose and trehalose, several less common lyoprotectants are used in research. Oligosaccharides, such as hydroxypropyl-β-cyclodextrin (HPβCD), are also used as lyoprotectants. In some studies, HPβCD has been shown to be as effective as both trehalose and sucrose. The results of van den Hoven et al. showed that the particle size of freeze-dried liposomes (107 nm) was essentially the same as their initial size prior to lyophilization (112 nm) [104]. Furthermore, less than 1% of the encapsulated drug experienced leakage during the lyophilization process, outperforming both trehalose and sucrose in the same study. Despite these encouraging results, however, the mass of data across many studies on the subject generally support trehalose as the premier lyoprotectant.

### 6.2. Process Parameters

The three stages of lyophilization have already been described. However, it is worth discussing how the parameters dictating these stages can have an impact on the finished product. During the freezing stage, water converts to ice as the chamber cools, leaving behind a solution of solute (lyoprotectant). This concentration increases until a temperature is reached at which the solution cannot become more concentrated, and no more water freezes. This is known as the glass transition temperature (Tg’), as described in Section 3.1. In order to ensure complete freezing for this stage, the temperature must be brought to and held below Tg’. However, the rate at which the system reaches that temperature could have an impact on the quality of the lyophilized product. Whether a slow cooling rate versus a quick cooling rate is advantageous for product quality remains somewhat controversial. Many papers claim that a quick cooling rate resulted in better quality of the lyophilized product [105,106]. The justification for this result is that a quicker cooling rate leads to finer ice crystals, which are less likely to damage the nanoparticles [107]. However, several papers also report a slow freezing rate having provided optimal results [94,108]. Howard et al. showed that when lipid nanoparticles were frozen quickly at −80 degrees Celsius the particle size increased by over 20%, whereas freezing the same particles slowly at −20 degrees Celsius resulted in a size increase of just 12.4%. The explanation for this result could be that a slower cooling rate reduces the osmotic pressure because the water molecules have time to diffuse across the membrane of the nanoparticle into the solution [109]. This reduces the likelihood of ice crystals forming inside the nanoparticle, which could otherwise cause irreversible damage. All things considered, there is still research to be conducted with respect to the freezing rate.

The primary drying stage is characterized by the sublimation of the frozen ice at low temperature and pressure. In order to induce sublimation, however, the temperature must be increased slightly. This is the point at which a critical error can occur. It is imperative that the temperature of the system not exceed the collapse temperature (Tc) of the product [7]. Otherwise, the porous structure of the lyophilized cake will collapse, making it far more difficult to resuspend the nanoparticles and keep residual moisture at low levels [110].

Secondary drying aims to eliminate the remaining water in the sample that did not convert to ice during freezing. The pressure remains near a vacuum while additional heat is added to the system to clear the adsorbed water from the sample. For the best long-term stability, pharmaceutical products at the end of this process should have a residual moisture content of less than 1% [111]. Although that is a clear, defined goal, achieving less than 1% moisture content is somewhat complicated. The optimal time and temperature for secondary drying is still unknown, and it is usually decided on a case-by-case basis through trial and error. Efforts to improve the efficiency of this process have been attempted, and many studies have worked towards creating a predictive model of secondary drying parameters [112,113,114]. However, at the time of this review, there is no agreed upon gold standard for secondary drying conditions.

### 6.3. Size of Nanoparticle

Although the success of lyophilization can affect the size of the nanoparticle (Section 5.1), the size of the nanoparticle can also affect the success of lyophilization. For example, it has been reported that liposomes in the range of 50–100 nm showed almost complete retention of their entrapped solute, whereas smaller (25 nm) and larger (200–400 nm) liposomes leaked 50–60% of the entrapped solute [115]. As mentioned previously (Section 5.3), free drug remaining in the sample during lyophilization can result in a decrease in absolute zeta potential [74], which leads to possible aggregation or particle fusion. Therefore, the size alone of specific nanoparticles holds an innate lyoprotective effect.

### 6.4. Encapsulated Drug

Another factor that plays a role in the stability of nanoparticles during lyophilization is the hydrophobicity of the encapsulated drug. A lipophilic drug is likely to be entrapped within the lipid-based portion of the particle, which for many nanoparticles is located in the membrane. A hydrophilic drug is likely to be entrapped in the aqueous pockets of nanoparticles, and an amphiphilic drug may be located in either site [116]. It has been reported that, as a result of these tendencies, hydrophilic and amphiphilic drugs are far more likely to experience leakage during lyophilization compared to lipophilic drugs. The results of Guimarães et al. show this effect very clearly. They investigated the leakage of methotrexate, doxorubicin, and tamoxifen from liposomes as a result of lyophilization. Methotrexate and doxorubicin were stored in the aqueous center of liposomes due to their hydrophilicity, and they experienced leakages of about 61.1 and 24.5%, respectively. Meanwhile, tamoxifen was stored in the lipid bilayer due to its lipophilicity and experienced only 4.0% leakage [117]. This stark difference highlights the effects of the drug itself on formulation stability during lyophilization. It is worth mentioning that drugs with a wide variety of lipophilicities have been successfully encapsulated and lyophilized in nanoparticles. However, prior knowledge of the chosen drug’s properties, such as lipophilicity, will aid in designing a robust process.

With all the above factors considered, it is therefore shown that lyophilization may be a potential solution to overcoming the stability challenges of nanoparticles. Table 1 summarizes examples of current FDA-approved lyophilized nanoparticles that are currently on the market with their classifications and indications (Table 1).

## 7. Alternatives to Lyophilization

Lyophilization is largely considered the gold standard in drying pharmaceutical products where stability is a major concern. However, the drying of nanoparticle formulations is a crucial step in the pharmaceutical production process, and therefore, several different methods exist. Two of the most common alternatives are described here.

### 7.1. Spray Drying

Spray drying is a commonly used drying technique across many different fields, and it is frequently used for pharmaceuticals to coat or encapsulate an active ingredient. It works by injecting a liquid solution through a nozzle that sprays the liquid at high velocities in the presence of hot air [119]. The liquid solvent is evaporated, and the result is a fine, solid powder. Spray drying has been frequently used to produce nanoparticles for various drug delivery routes, especially for inhalation [120]. It has been shown to be reproducible, affordable, quick, and easily scalable [121]. As a result, several FDA-approved drug substances are encapsulated using this technology [122]. However, a potentially very low yield remains a significant hurdle, as the process may only allow for a 20–70% recovery [121]. Additionally, the compatibility of heat-sensitive therapeutics, such as protein and lipid nanoparticles, may be a concern depending on the specific molecule and operating conditions. The formulation’s exposure to high temperatures during the spray-drying process can lead to a change in protein structure and activity [123]. Despite this, studies have shown that with the optimization of key parameters such as inlet temperature, flow rate, and formulation concentration, sensitive nanoparticles can be spray dried successfully. For example, Lee et al. showed that with optimal operating conditions, bovine serum albumin could be spray dried while achieving a yield of over 70% [124]. All in all, spray drying holds several advantages for increasing the long-term stability of a drug substance, but its low yield and its uncertainty surrounding heat-sensitive therapeutics are a major cause of concern for nanoparticle formulations.

### 7.2. Electrospraying

Electrospraying is a drying method that utilizes electrostatic forces to cause the atomization of an injected liquid solution. It is used to encapsulate an active ingredient in a polymer solution, and it is known for being one of the most efficient technologies for the production of nanoparticles [119]. In addition to encapsulation efficiency, electrospraying also allows for modular process parameters that can be adjusted according to the desired characteristics of the drug product [120]. Furthermore, the process takes a fraction of the time required for lyophilization, and it can be completed in just one simple step [119]. However, the high efficiency and quick speeds are not enough to compensate for the incredibly slow flow rate (microliters per minute) that is characteristic of this process [121]. As a result, electrospraying would be both a logistical and economic nightmare to scale up to a level that is industrially relevant. For this reason, electrospraying is currently only feasible on a small scale, such as in a laboratory.

## 8. Conclusions and Future Perspectives

Nanomedicine has found its place in pharmaceuticals over the last 40 years, and as it continues to expand its utility, it does not appear to be going anywhere. However, there is much research needed to catapult nanoparticles into the forefront of new drug development. There are legitimate concerns regarding the long-term stability of many nano-formulations that warrant further investigation.

Although lyophilization has been proven to ameliorate these issues, there are still many process parameters that need to be optimized in order to improve the final result. The ideal lyoprotectant mixtures, initial freezing rates, and secondary drying parameters remain significant gaps in our understanding of lyophilization. In addition to these barriers, the wide variety of structures and compositions of existing nanoparticles make it unlikely that one common protocol will ever be sufficient.

With that being said, of the currently known drying methods, lyophilization does appear to offer the greatest chance at achieving long-term stability for many nanomedicines. Shelf life is significantly increased, making storage and transportation a much safer process for drug-loaded nanoparticles. In addition, current characterization techniques make evaluating the quality of lyophilized nanoparticles a robust process. For these reasons, lyophilization will likely continue to be utilized and improved upon for the long-term storage of therapeutic nanoparticles.

## Figures and Tables

**Figure 1 ijms-24-14041-f001:**
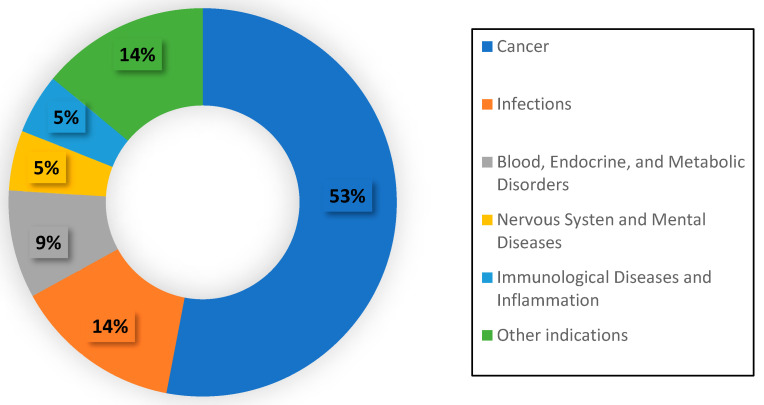
Pie chart illustrating the breakdown of indications of approved nanoparticles.

**Figure 2 ijms-24-14041-f002:**
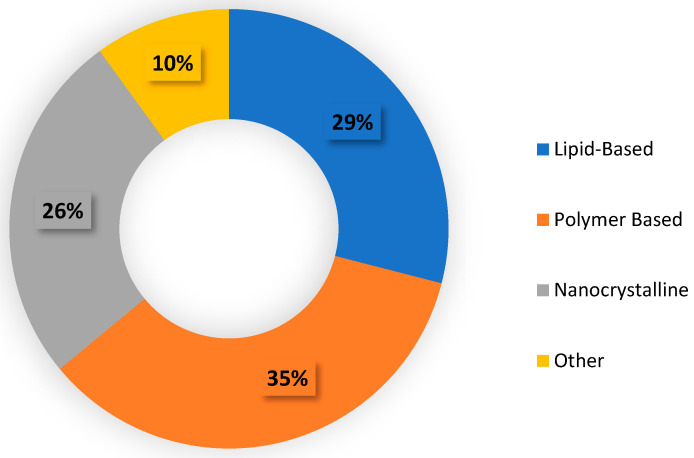
Pie chart illustrating a breakdown of the different approved nanoparticle types.

**Figure 3 ijms-24-14041-f003:**
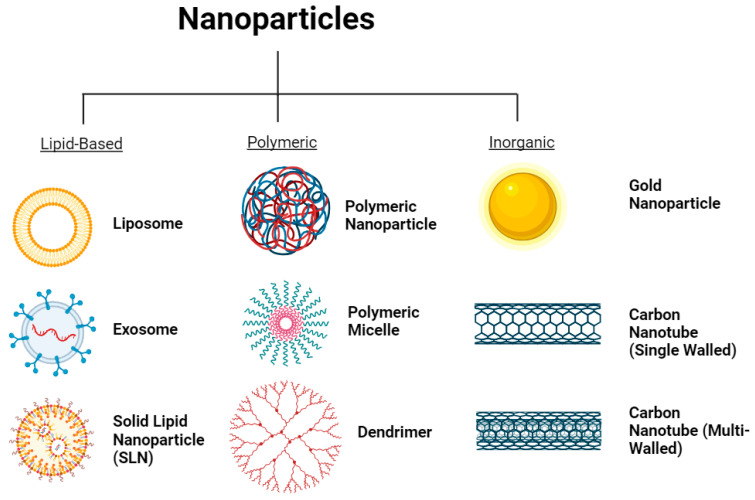
Major classes of nanoparticles.

**Figure 4 ijms-24-14041-f004:**
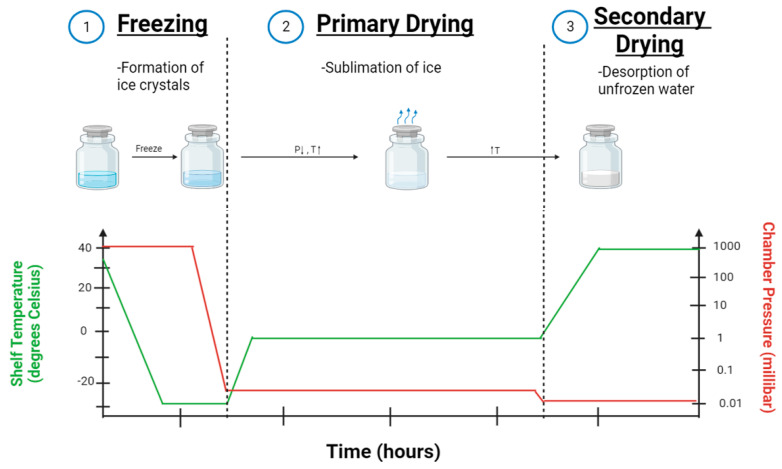
A schematic of parameters and sample states throughout the lyophilization process.

**Figure 5 ijms-24-14041-f005:**
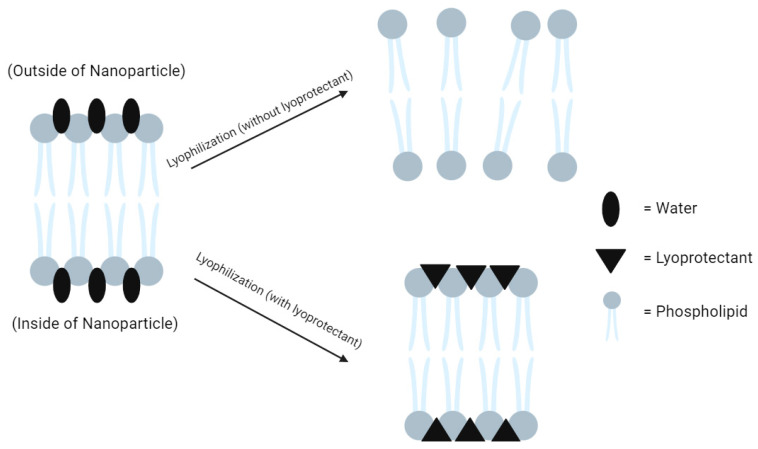
Schematic illustrating the water replacement theory.

**Table 1 ijms-24-14041-t001:** Examples of FDA-approved lyophilized nanoparticles.

Marketed Name	Sponsor	Nanoparticle Type	Indication	Reference
Ambisome	Gilead Sciences	Liposomal	Fungal infections	[118]
Visudyne	Bausch and Lomb	Liposomal	Macular degeneration	[118]
Cimzia	UCB	Polymeric	Crohn’s disease, arthritis, psoriasis	[118]
Somavert	Pfizer	Polymeric	Acromegaly	[118]
Ryanodex	Eagle Pharmaceuticals	Inorganic	Malignant hyperthermia	[118]
Abraxane	Celgene	Protein–drug conjugate	Breast, lung, and pancreatic cancer	[118]

## Data Availability

Not applicable.

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
