# Peer review of "Lyophilization of Nanoparticles, Does It Really Work? Overview of the Current Status and Challenges"

_ijms, 2023, doi:10.3390/ijms241814041_

Round 1
Reviewer 1 Report
The topic of this manuscript is important and current. The work is well written with the support of current literature. However, changes have to be entered into the revised version of the manuscript before it can be further processed:
1. chapter 2.2 - more details on mechanisms to improve individual properties should be added. Why is solubility improved? What it comes from?
2. In the content of the entire work, I miss references to literature data. I see a lot of references to the literature, but few numeric data and specific impact on a given parameter (particularly in chapter 6)
Author Response
- “Chapter 2.2 - more details on mechanisms to improve individual properties should be added. Why is solubility improved? What it comes from?.”
Response: We thank the reviewer for the positive feedback and helpful suggestions. We have updated section 2.2 to include these changes.
- “In the content of the entire work, I miss references to literature data. I see a lot of references to the literature, but few numeric data and specific impact on a given parameter (particularly in chapter 6)”
Response: We agree with this suggestion and thank the reviewer for the feedback. Data-based references have been added in Sections 6.1, 6.2 and 6.4
In summary, we have considered all suggestions from the reviewer, and we appreciate the positive feedback. We feel we have made the necessary updates to the manuscript, and we hope they are satisfactory for publication to IJMS.
Author Response
- “Which nanoparticles? Give some examples..”
Response: We thank the reviewer for the feedback. Specific examples have been added to the manuscript.
- “Please rephrase the sentence.”
Response: We agree that the sentence needed restructuring, and the manuscript has been updated to reflect this.
3.“Researchers use oligosaccharides also as a lyoprotectant e.g. b-hydroxypropyl cyclodextrin. Provide some examples”
Response: We thank the reviewer for the positive feedback, and we have updated the manuscript with a section on oligosaccharides as lyoprotectants.
4.“What about heat sensitive drug formulations; like lipid, protein, or protein conjugated polymeric NPs? Is spray drying suitable method for such formulations? Explain.”
Response: We appreciate the reviewer’s suggestion, and we agree. A section on spray drying’s effect on heat-sensitive nanodrug formulations has been added to the manuscript.
In summary, we have considered all suggestions from the reviewer, and we appreciate the positive feedback. We feel we have made the necessary updates to the manuscript, and we hope they are satisfactory for publication to IJMS.
Reviewer 3 Report
The manuscript represents a review on a very important topic that has not yet been sufficiently covered in the literature. Lyophilization of nanoparticles and nanoparticle-based delivery systems is of great practical interest, but at the same time it can face many obstacles. The manuscript is well written and structured, the number of figures and tables is sufficient. Overall, the manuscript merits to be published in IJMS.
Minor:
Why did the authors mention only gold NPs and carbon nanotubes out of the whole variety of inorganic nanoparticles? It would be better to make the following division:
2.3.3. Inorganic nanoparticles
1) Metal nanoparticles (gold, silver, etc)
2) Magnetic nanoparticles
3) Quantum dots
2.3.4. Carbon nanoparticles (carbon nanotubes, fullerenes, and graphene derivatives)
Author Response
1.“ Why did the authors mention only gold NPs and carbon nanotubes out of the whole variety of inorganic nanoparticles? It would be better to make the following division:
2.3.3. Inorganic nanoparticles
1) Metal nanoparticles (gold, silver, etc)
2) Magnetic nanoparticles
3) Quantum dots
2.3.4. Carbon nanoparticles (carbon nanotubes, fullerenes, and graphene derivatives)
Response: We appreciate the reviewer’s positive feedback. We have addressed the suggestion in the updated manuscript in section 2.3.
In summary, we have considered all suggestions from the reviewer, and we appreciate the positive feedback. We feel we have made the necessary updates to the manuscript, and we hope they are satisfactory for publication to IJMS.
Round 2
Reviewer 1 Report
Accept in present form